# Factors Affecting Response Rates of the Web Survey with Teachers

**Konstantinos Lavidas** [1,*], **Antonia Petropoulou** [1], **Stamatios Papadakis** [2], **Zoi Apostolou** [1], **Vassilis Komis** [1], **Athanassios Jimoyiannis** [3] **and Vasilis Gialamas** [4]

1   Department of Educational Sciences and Early Childhood Education, University of Patras, 26504 Patras, Greece
2   Department of Preschool Education, Faculty of Education, University of Crete, 74100 Crete, Greece
3   Department of Social and Educational Policy, University of Peloponnese, 20100 Korinthos, Greece
4   Department of Early Childhood Education, National and Kapodistrian University of Athens, 11527 Athens, Greece
*   Correspondence: lavidas@upatras.gr

**Abstract:** Although web survey has been a popular method of data collection in the academic community, it presents meagre response rates, which primarily affect the validity of the results as well as the reliability of the outcomes. Surveys worldwide that study the response rate only of teachers have not been found in the relevant literature. In this survey, with a sample of 263 Greek teachers, we investigate possible factors that explain teachers' intention to participate in web surveys that are conducted by online questionnaires indicating, therefore, the factors that probably influence the response rate of web surveys. Our findings support those factors such as (a) authority, (b) incentives, (c) survey structure/form, (d) ethical issues, (e) reminders and pre-notifications, and (f) survey time received, which seem to explain the teachers' intention to participate in web surveys with questionnaires. Based on the findings, methodology implications and limitations for researchers are discussed.

**Keywords:** web survey; online survey; internet-based survey; questionnaire study; teachers' response rate; educational research; completeness of data

## 1. Introduction

Last two decades, web or internet surveys have been a popular method of data collection in the academic community [1,2]. Web surveys are being conducted in web-based environments where the potential participants that have been invited can find and fill out an online questionnaire. Web survey offers several advantages such as easy access and participants' anonymity [3], quicker response, time and cost-saving method, simple processing and fast storing of data and the ability to collect data from large population areas [3–9]. Due to these advantages, traditional forms of research are conducted by mail or telephone. Despite the advantages offered, web surveys present meagre response rates [6,8–13], which primarily affect the validity of the results and the reliability of the outcomes threatening the inferential value of the survey method [1,14–16]. This fact is important because much of the published academic research in influential journals is based on web surveys [17].

Considering the low response rate of web surveys, researchers focus on investigating the factors that may be related to this issue. In the last decade, they have conducted several studies on the factors that affect the survey response rate. Specifically, more of them investigated the factors that affect the response rate of web surveys (e.g., [3,7,18]), and fewer studies on the response rate both of web surveys and postal or paper surveys (e.g., [19,20]). Notably, researchers have identified the problem of low response rates in web surveys from various perspectives (see Section 2). However, relevant studies investigated some factors but not all of them overall. These factors are "consent to participate in the survey" [21], "location of demographic information in the questionnaire", "demographics and other

personal information discourage responders from participating", "mandatory answers", and "a reference to the survey's precise purposes" [2], option to choose an answer: "I do not know/I do not answer" [22], "time limit (deadline)" [9,23,24], and "approval of ethics committee". In the present study, we attempt to present the factors influencing the response rate by doing an extensive literature review.

Additionally, most previous research has been conducted in countries outside Europe. Specifically, they have been done in countries such as the United States of America [1,2,4,7,13,20,22,25–27], Japan [15], and Chile [23]. Only six of them have been conducted in Europe and specifically in Denmark [8,19], German [9], Spain [3], Slovenia [24] and Sweden [18], while no relevant research has been done in Greece.

Moreover, these surveys mainly concern participants without being referred to a particular cohort [8,18,19,24,26], graduate and undergraduate Students [1,2,13,19,28,29], primary care health professionals [3], radiologists [27], physician specialist [25], and only three concern educators and specifically University faculty [4] and school principals [20,23]. Similar surveys worldwide that only study teachers' response rates have not been found in the relevant literature.

Therefore, in this survey, with a sample of Greek teachers, we investigate a wide range of possible factors that explain teachers' participation in web surveys that are conducted by online questionnaires. This study has significant implications since teachers are frequently requested to complete surveys. Therefore, we hope to enrich the existing bibliography by providing empirical insights into the factors that explain the response rate of web surveys in education.

## 2. Literature Review

Increasing participation rates in web surveys are challenging for researchers [20]. The international literature has presented several factors that explain the willingness of individuals to participate in a survey. The factors that researchers demonstrate are (a) incentives, (b) authority, (c) survey structure/form, (d) ethical issues, (e) pre-notification and reminders, and (f) survey time received.

Specifically, the factor "insensitive" refers, on the one hand, to "external incentives" such as small financial incentives [1,7,13,20,25–27,29,30] or donation to charities incentives [8]. "Monetary incentives" are one of the more critical factors that can increase the response rate on a survey [20]. This is evidenced by Smith et al.'s [26] study, where in order to increase the participation rates in the research, he included in the invitation a sum of $2. Indeed, the results of his research showed that prepaid monetary incentives were an effective method of improving response rates. On the other hand, "internal" incentives such as interest in the research subject [8,9,25] or the participant attitudes in an investigation [2,10]. The category "Attitude towards research" also includes the generous incentives influencing a responder to participate in research. An altruistic motivation can be considered when the respondents take part in the research as they consider that their answers can contribute to the achievement of a good purpose or just to take part in helping with their participation in the research [22,24]. "Interest" in the subject of the research is an essential factor for increasing the response rate for any research. Indeed, as Park et al. [7] confirmed, respondents are much more likely to participate in a survey when its topic is related to their interests.

The sponsor or "Authority" of the research seems to explain the willingness of someone to participate in a web survey [2,7,24,28,29]. When the sponsor is a prestigious agency, such as a governmental agency or an educational agency, and not a commercial one, participants are more likely to feel more secure in trusting their personal information and data [24]. Even though researchers often point it out, some [2,7] argue that it does not significantly influence the participants' willingness to complete a survey. Therefore, it would be of particular research interest to examine it further.

The "survey structure/form" concentrates on factors such as personalization of invitations, open-ended/close-ended questions, and questionnaire length. In more detail, the

factor "Personalization of invitations" [7,13,22,25,29,31] with a positive effect on response rates, mainly in web surveys [13,25]. Response rates could be increased when the researcher uses personalization tactics, such as email invitations that include the respondents' first or last names in the subject's title [22]. The "length of the survey" [7,22,26,27,29] could influence a person's participation decision. Researchers conclude that the less time a questionnaire takes to be completed, the more likely it is to be completed by the respondent. Surveys lasting more than 13 min present lower response rates [22] than surveys lasting less than 10 min [7]. Saleh and Bista [2] also indicate in their research that the form of questions, when they are open-ended, discourages the respondents from completing the survey. In this context, they suggest using close-ended questions to achieve higher response rates to their surveys.

"Ethical issues" such as consent for participation, data safety, privacy, and anonymity, although they have scarcely been studied (e.g., [21]), seem to explain the response rate of the web surveys. According to Nayak and Narayan [21], it is a critical issue of ethical surveys to give the respondents the choice of consent to participate in a survey. Also, researchers should assure the anonymity of the participants since data safety and privacy has been pointed out by researchers that have a strong correlation with the high response rates [2].

"Reminders and pre-notification" [1–3,9,22,25,31] are considered essential factors since they could increase the response rate. Specifically, on the one hand, "pre-notification" can provide some information and invitation calls for the participants to join the survey [10]. On the other hand, "reminders" act as an additional tool in case the potential respondents do not initially receive the invitation or have not studied the necessary information from the first invitation [10]. Indeed, the correct number of "reminders" to avoid becoming annoying to the participants is at the centre of research interest [25].

"Survey time received" is identified in the literature that explains the response rates to web surveys [23]. Specifically, Madariaga et al. [23] mentioned that scheduling when the survey will be sent to the respondent for completion is a factor that can affect the response rate considering some groups of citizens, such as school principals, have increasingly busy daily schedules. However, this factor needs further examination as Saley and Bista [2] did not find similar findings.

## 3. Objective of Research

The previous Studies include specific factors that cover part of the explanation of the web survey response rate or have controversial results. This study attempts to give us a clearer and broader image, including the previous factors and others that have scarcely been investigated. Moreover, the study's novelty lies in the fact that this study investigates the factors that explain the response rate of web surveys for the population of teachers. Consequently, the present research aims to investigate whether specific factors correlate with the teachers' intention to participate in a web survey with a questionnaire. Therefore, we will investigate the factors that probably explain the web survey response rate. The investigated factors are authority, incentives, survey structure/form, ethical issues, reminders and pre-notifications, and survey time received.

## 4. Methodology

### 4.1. Research Procedure

In this research, we followed a cross-sectional quantitative research method [32], and the web questionnaire was administered in November 2021. This research was approved by the Research Ethics Board designated by the Department of Educational Science and Early Childhood Education of the University of Patras. Specifically, 400 randomly selected participants from a list of teachers were invited via email to participate in the web survey, voluntarily and anonymously, without any monetary incentive. The email message contained a cover letter and a link. In the cover letter, participants could read (a) the aim of the survey, (b) insurance for the assurance of the participants' anonymity, (c) analytic

information and guidelines about the survey, and (d) a choice where the participants should declare their consent in the research. Google form was utilized as an online survey collection tool. The participants spent about eight minutes completing the questionnaire. Finally, the participants received no reminders after the initial administration of the questionnaire.

*4.2. Research Instrument*

In order to create the research instrument, we relied on Park et al. [7], Saleh and Bista's [2] and Koskey et al. [22] results of their surveys. We modified the questions to respond to the requirements of the present survey by including items from all the explained factors of response rate as described in the literature review. The web questionnaire consisted of two sections. The first section included five items related to demographic variables (gender, age, educational stage, postgraduate studies, research experience), and the second section included 31 items which were the main statements related to factors affecting response rate on web surveys. One of them is the teacher's intention to participate in a web survey with a web questionnaire, while the others 30 correspond to the factors we identified in the literature. In the tables of the Section 7, you can see the 30 items. More precisely, the web questionnaire included the following group of items: authority (3 items) [2,7], incentives (6 items) [2,7,22], survey structure/form (12 items) [2,7,22], ethical issues (6 items) [2] and reminders and pre-notification (3 items) [2,7,22]. There are 30 close-ended questions and one open-ended question. The main responses to the closed-ended items were indexed on a 4-point Likert-type scale (1. strongly disagree, 2. disagree, 3. Agree, and 4. strongly agree). All the 31 items are presented in the Appendix A.

In order to enhance the content validity of the research instrument [32], four researchers working in the field of methodology of educational research evaluated whether the above items fit in each of the above factors. For the assessment procedure, a 3-point scale was used (not relevant, relevant, very relevant). According to this evaluation process, all items were kept in the final web questionnaire since no one item was not evaluated as not relevant. Subsequently, the web questionnaire was initially piloted in 10 teachers who were not included in the final sample, asking them to email their comments and suggestions apart from completing the questionnaire. Considering their responses and feedback, we made some revisions to improve the wording of some questions.

Finally, to support the stability of this version of the research instrument over time, a test-retest process was performed on 15 teachers who were not included in the final sample. The assessment of the similarity of answers with the Wilcoxon signed-rank nonparametric test [33] did not reveal statistically significant differences (see Table A1 in the Appendix A) among the paired items in the two phases (a week interval).

## 5. The Strategy of Data Analysis

The statistical software SPSS version 24.0 was utilized for data management and statistical analyses [33]. Initially, a descriptive analysis was performed to explore teachers' views about the possible factors that affect them to participate in a web survey with a questionnaire. Subsequently, to test whether the previous factors probably affected the teachers' intention to participate in a questionnaire web survey, we calculated non-directional and directional measures of association, the Goodman-Kruskal gamma ($\gamma$) and the Somers'd coefficients, respectively. Goodman-Kruskal gamma coefficient reveals the strength of association between ordinal variables [34]. However, this measure does not distinguish between dependent and independent variables. To assess the dependent variable's strength and degree of dependency (teacher's intention to participate in a web survey with a web questionnaire). Somers'd (delta) coefficient was utilized. This is a suitable coefficient of the strength and the direction of association between an ordinal dependent variable and an ordinal independent variable [35]. Both coefficients are ranged [−1 to 1]. When the values are at least 0.1 or above, the relationship strength between two variables is considered not negligible [36].

## 6. Participants

We called 400 randomly selected participants from a list of teachers from western Greece, and from them, two hundred and sixty-three (263) teachers participated in the study. So, the response rate is 65.75% (263/400). This high response rate creates a more substantial claim in generalizing results from the sample to the population [32]. Table 1 presents participants' demographic information.

**Table 1.** Demographical information of participants (*N* = 263).

|  | Frequency | Percent |
|---|---|---|
| Gender |  |  |
| Male | 55 | 20.9 |
| Female | 208 | 79.1 |
| Age |  |  |
| ≤30 | 36 | 13.7 |
| 31–35 | 23 | 8.7 |
| 36–40 | 27 | 10.3 |
| 41–45 | 36 | 13.7 |
| 46–50 | 49 | 18.6 |
| 51–55 | 63 | 24.0 |
| ≥56 | 29 | 11.0 |
| Educational stage |  |  |
| Preschool | 87 | 33.1 |
| Primary | 104 | 39.5 |
| Gymnasium | 28 | 10.6 |
| High school | 44 | 16.7 |
| Postgraduate Studies |  |  |
| No | 79 | 30.0 |
| Yes | 184 | 70.0 |
| Research Experience |  |  |
| No | 143 | 54.4 |
| Yes | 120 | 45.6 |

## 7. Results

To have a more cohesive presentation of the teachers' responses, we presented analytically in all the tables the percentages of "disagree" and "strongly disagree" as well as "agree" and "strongly agree". Also, in each table, we present the Goodman-Kruskal gamma ($\gamma$) coefficients, a measure of association between teachers' intention to participate in a web survey with a questionnaire and all possible factors that explain this behaviour. Also, the statements were sorted in descending order according to the last column (i.e., items with the highest association coefficient appear at the top).

Regarding the teachers' intention to participate in a web survey with a questionnaire (see Appendix A. I will participate in a web survey with a questionnaire), almost all participants answered that they agree (67%) or strongly agree (30%). Only 3% of them disagree or strongly disagree. Pearson chi-square results do not indicate significant associations between the teachers' intention and all the demographic variables: gender ($\chi^2(2) = 0.001$, $p = 0.99$), group of age ($\chi^2(12) = 12.218$, $p = 0.43$), educational stage ($\chi^2(6) = 4.936$, $p = 0.55$), postgraduate studies ($\chi^2(2) = 2.562$, $p = 0.28$), and research experience of teachers ($\chi^2(2) = 0.232$, $p = 0.89$).

Relate to the authority conducting the web survey (Table 2). Participants presented a high percentage of positive responses (agree to agree strongly) to the statements related to this factor. However, the Goodman-Kruskal gamma showed moderate positive significant associations of conducting web surveys only by a prestigious organization and by known colleagues with teachers' intention to participate in the survey.

**Table 2.** Teachers' response percentage frequencies on the statements of Authority factor and association coefficients with teachers' intention to participate in a web survey with a questionnaire.

| | nr | SD | D | A | SA | γ | d |
|---|---|---|---|---|---|---|---|
| 2. I complete research conducted by prestigious organizations (Universities, Government agencies, Public agencies, etc.). | 0.8% | 2.3% | 17.1% | 46.4% | 33.5% | 0.344 ** | 0.163 * |
| 4. I fill out a web questionnaire conducted by a person I know (e.g., teacher, colleague, friend). | 0.4% | 1.1% | 6.5% | 35.7% | 56.3% | 0.318 ** | 0.163 * |
| 3. I complete a web questionnaire conducted by an organization I belong to (or if my institution sponsored the survey). | 1.1% | 1.9% | 10.6% | 49.0% | 37.3% | 0.189 | 0.088 |

Note: nr = no response, SD = Strongly Disagree, D = Disagree, A = Agree, SA = Strongly Agree, γ = Goodman-Kruskal gamma, d = Somer's coefficient, * $p < 0.01$, ** $p < 0.001$.

Regarding the incentives that motivate participants to complete a survey (Table 3), teachers stated that they mainly agree or strongly agree with the four first statements related to internal incentives and lower degree with the last two statements related to external motivations. Similarly, the two coefficients confirm that their internal incentives mainly motivate them to participate in a web survey.

**Table 3.** Teachers' response percentage frequencies on the statements of Incentives factor and association coefficients with teachers' intention to participate in a web survey with a questionnaire.

| | nr | SD | D | A | SA | γ | d |
|---|---|---|---|---|---|---|---|
| 7. I participate in a web survey to benefit society by helping inform research; I generally see value in research efforts. | 1.9% | 0.4% | 4.9% | 53.6% | 39.2% | 0.486 ** | 0.242 ** |
| 10. I participate in a web survey with a questionnaire when the subject indicates the research purpose. | 1.5% | 0.4% | 1.9% | 40.3% | 55.9% | 0.450 ** | 0.209 ** |
| 8. I complete a survey with a web questionnaire if I am interested in the topic. | 0.8% | 0.4% | 6.1% | 42.2% | 50.6% | 0.390 ** | 0.190 ** |
| 9. I participate in a survey with a web questionnaire if the researcher promised to inform me about the findings | 2.7% | 1.5% | 25.1% | 51.0% | 19.8% | 0.382 ** | 0.184 ** |
| 5. I participate in a survey with a web questionnaire if only I am promised a monetary reward. | 1.1% | 29.7% | 42.2% | 12.5% | 14.4% | 0.069 | 0.032 |
| 6. I participate in a survey with a web questionnaire if I am aware that my participation will offer an amount as a donation to an organization. | 4.6% | 4.6% | 23.6% | 37.3% | 30.0% | 0.034 | 0.016 |

Note: nr = no response, SD = Strongly Disagree, D = Disagree, A = Agree, SA = Strongly Agree, γ = Goodman-Kruskal gamma, d = Somer's coefficient, ** $p < 0.001$.

About the Survey's structure/form (Table 4), a high percentage of teachers declared that they mainly prefer to complete questionnaires when it mainly includes concise, closed-ended questions, the option "I do not know and/or I do not answer" when they know in advance the time required to complete the questionnaire and the number of questions. Parallelly, according to the two coefficients, these factors correlate with the participants' intention to participate in a web survey. However, the professional look of the questionnaire, the mandatory answers, and the inclusion of participants' names do not present a significant association with teachers' intention to participate in the survey.

**Table 4.** Teachers' response percentage frequencies on the statements of Survey structure/form factor and association coefficients with teachers' intention to participate in a web survey with a questionnaire.

| | nr | SD | D | A | SA | γ | d |
|---|---|---|---|---|---|---|---|
| 16. I participate in a web survey with a questionnaire if only the questions items are petite and summarizing | 0.4% | 0.4% | 0.4% | 41.4% | 57.4% | 0.712 ** | 0.354 ** |
| 18. I participate in a web survey with a questionnaire if only the questions are closed-ended (including all possible answers). | 1.9% | 1.1% | 4.2% | 46.0% | 46.8% | 0.609 ** | 0.313 ** |
| 19. I participated in a web survey with a questionnaire if only the questions included "I do not know and, or I do not answer" options. | 0.4% | 0.8% | 10.6% | 60.5% | 27.8% | 0.616 ** | 0.329 ** |
| 12. I participate in a survey with a web questionnaire if I know how long it will take to fill out beforehand. | 0.8% | 1.5% | 7.6% | 54.4% | 35.7% | 0.474 ** | 0.226 ** |
| 13. I participate in a survey with a web questionnaire if I know the number of questions. | 2.3% | 2.7% | 17.1% | 51.3% | 26.6% | 0.306 ** | 0.141 * |
| 17. I participate in a web survey with a questionnaire if only the question items are open-ended (not all possible answers are included, and text is needed). | 2.3% | 23.2% | 61.2% | 11.4% | 1.9% | −0.312 ** | ·0.159 * |
| 15. If the email invitation looks professional, I participate in a web survey with a questionnaire. | 4.9% | 1.5% | 22.1% | 52.5% | 19.0% | 0.123 | 0.059 |
| 11. I fill out a web questionnaire when the answers are not mandatory. | 6.1% | 3.8% | 28.9% | 49.4% | 11.8% | 0.079 | 0.039 |
| 14. I participate in a web survey with a questionnaire if the invitation includes my name. | 6.8% | 11.4% | 50.2% | 22.8% | 8.7% | 0.023 | 0.011 |

Note: nr = no response, SD = Strongly Disagree, D = Disagree, A = Agree, SA = Strongly Agree, γ = Goodman-Kruskal gamma, d = Somer's coefficient, * $p < 0.01$, ** $p < 0.001$.

About the time required to complete a questionnaire (see Appendix A with possible answers: less than 5′, 6–10′,11–15′, 16+), overall, 85.9% of the sample participants stated that they prefer to complete questionnaires when the total time does not exceed the 10 min. Regarding the deadline for completion of a questionnaire (see Appendix A, I participated in a web survey with a questionnaire if the deadline is: 1 week, two weeks, three weeks, or four weeks), 61.7% answered up to 1 week, the 24.5% answered up to 2 weeks, and the remaining teachers answered up to 4 weeks. Regarding the place of demographic information in a questionnaire (see Appendix A with possible answers: at the beginning, at the end, not to exist), most of the sample (87.5%) answered that they prefer it to be at the beginning of the questionnaire.

Regarding ethical issues related to research (Table 5), teachers mainly agree or strongly agree that these issues determine their participation in a web survey. This is confirmed by the significant correlations of these factors with teachers' intention to participate in the survey.

**Table 5.** Teachers' response percentage frequencies on the statements of Ethical issues factor and association coefficients with teachers' intention to participate in a web survey with a questionnaire.

|  | nr | SD | D | A | SA | $\gamma$ | d |
|---|---|---|---|---|---|---|---|
| 27. I participate in a web survey with a questionnaire if this research has received the approval of the research ethics committee. | 8.0% | 1.1% | 11.0% | 52.9% | 27.0% | 0.427 ** | 0.211 ** |
| 24. I participate in web surveys with a questionnaire if I am assured of my anonymity | 2.3% | 0.0% | 5.7% | 39.2% | 52.9% | 0.417 ** | 0.196 ** |
| 25. I participate in web surveys with a questionnaire that I know secures my data and information. | 0.4% | 0.4% | 2.7% | 22.4% | 74.1% | 0.409 ** | 0.176 * |
| 23. I fill out a web questionnaire when asked about my consent. | 4.2% | 1.5% | 12.5% | 53.2% | 28.5% | 0.331 ** | 0.164 * |
| 26. I would like to know how the researcher accessed my account when the registration invitation was sent via email. | 2.3% | 1.1% | 5.7% | 38.8% | 52.1% | 0.304 ** | 0.142 * |

Note: nr = no response, SD = Strongly Disagree, D = Disagree, A = Agree, SA = Strongly Agree, $\gamma$ = Goodman-Kruskal gamma, d = Somer's coefficient, * $p < 0.01$, ** $p < 0.001$.

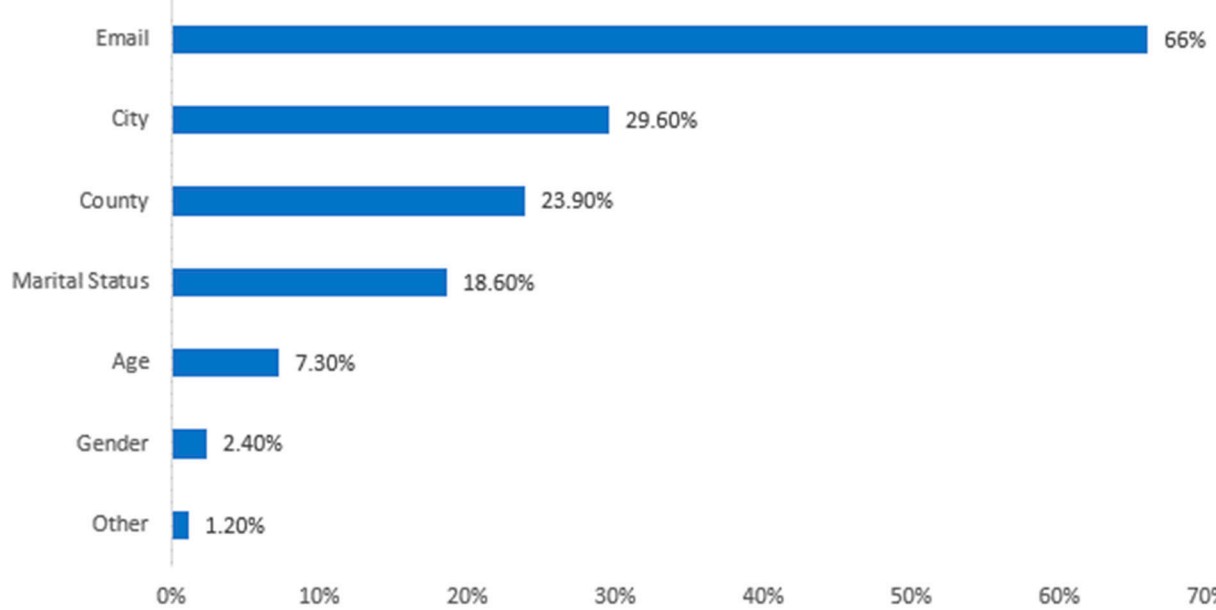

**Figure 1.** The demographic information discourages participants from completing a web survey.

About the reminders and pre-notification, teachers declared that they desire to complete a web survey (Table 6) when the invitation to participate is being sent by email and in the lower degree, when they see it on a website the invitation, or they receive a reminder to complete the survey. Similarly, the gamma coefficient revealed a significant positive correlation between the invitation to participate being sent by email with teachers' intention to participate in the survey.

**Table 6.** Teachers' response percentage frequencies on the statements of Reminders and pre-notification factor and association coefficients with teachers' intention to participate in a web survey with a questionnaire.

| | nr | SD | D | A | SA | γ | d |
|---|---|---|---|---|---|---|---|
| 29. I participate in a web survey with a questionnaire when the invitation to participate is sent by email. | 7.6% | 1.5% | 20.5% | 57.4% | 12.9% | 0.283 ** | 0.144 * |
| 30. I participate in a web survey with a questionnaire if I see on a website the invitation (link) | 7.6% | 8.0% | 43.3% | 34.6% | 6.5% | 0.101 | 0.050 |
| 31. I participate in a web survey with a questionnaire if only I receive a reminder to complete it. | 3.8% | 4.2% | 32.3% | 52.9% | 6.8% | 0.153 | 0.075 |

Note: nr = no response, SD = Strongly Disagree, D = Disagree, A = Agree, SA = Strongly Agree, γ = Goodman-Kruskal gamma, d = Somer's coefficient, *$p < 0.01$, **$p < 0.001$.

## 8. Discussion of Results

This research aimed to confirm whether specific factors correlate with the teachers' intention to participate in a web survey with a questionnaire and, therefore, to indicate the factors that probably influence the web survey research response rate. The discussion below is mainly based on the statements' statistically significant associations (gamma and delta coefficients) with the teachers' intention to participate in the web survey.

The findings reveal that the respondents are more likely to participate in the web survey when the survey sponsorship has a reputation or is known to the participants. These results support earlier findings (e.g., [2,10,28,29]). However, the results diverge from the similar surveys of Park et al. [7] with a sample of 540 undergraduate and graduate students and the study of Petrovcic et al. [24] with an extensive sample of 2500 responders from the online health community.

Regarding the factor incentives, the teachers seem motivated to participate in a web survey from internal incentives. The results converge with the survey of Petrovcic et al. [24], which recommended that a "plea for help" is an essential factor in response rate. Interest in the topic and content of the web survey also seems to be an essential factor that affects the response rate in web surveys. The results are in line with the existing research literature on the impact of the survey topic on the response rate [2,7,10]. Moreover, our survey results indicated two additional factors that affect the response rate on web surveys: the reference of the explicit purpose of the survey and the promise of informing about the results.

Results also indicated that paying attention to the questionnaire structure/form leads teachers to participate in a web survey. Participants declared that they were more likely to complete a survey if only the questions were simple, comprehensive closed-ended and when they knew in advance both the time required to complete it and the number of questions it contained. The results agree with the survey of Fan and Yan [10]. Moreover, most of the sample stated that they discourage completing web surveys when the questions are open-ended [2]. A factor that brought up the dimension "Survey structure/form" and seems to affect the response rate on web-survey are the options "I do not know and/or I do not answer". Common sense suggests that longer questionnaires will tend to yield lower response rates than shorter questionnaires, as they require more time from the respondents to complete them. Indeed, almost nine of ten sample participants stated that they prefer to complete web questionnaires when the total time for it is not more than 10 min and the deadline to complete it is up to 3 weeks. This finding agrees with the existing literature involving the impact of survey length on response rate (e.g., [7,22,37,38]). We also investigated the influence of the location of demographic information in a web survey as a factor influencing the response rate. The results showed that participants (9 out of 10) would prefer it to be at the beginning of the web questionnaire.

Our findings revealed that consent for participation, data safety, privacy, and anonymity are critical ethical issues in web surveys. These results seem to align with the existing literature on the ethical and moral issues governing research [2,10,21]. The results of our research highlighted two more factors that seem to explain the web-survey response rate. The first is the approval of the research from an official ethics commitment service, and the second is the disclosure to the participant of how the researcher gained access to his/her account, in case the survey invitation is sent by email. We also examined whether some demographic information discouraged participants from completing the survey, and the results showed that they were mainly discouraged when asked for their email addresses and city of residence.

Moreover, the results indicated that participants are more likely to complete a web survey when the invitation to participate is sent by email [37]. It is worth noting that emails, in most cases, are blindly sent (using Blind Carbon Copy) by researchers to the responders based on an email list and do not have a personal suggestion for each participant. So, this is not contrary to our findings, where most of the participants are denied completing a web survey when they should complete their email addresses. Therefore, this result seems to align with the "ethical issues" that govern a survey, as mentioned above. Also, the results demonstrated that the reminder to complete the questionnaire does not explain the response rate. This finding differs from earlier studies about reminders, demonstrating a positive effect on response rate [3,4,7,9,13,25].

Finally, something worth noting is that the impact of the pandemic on web survey responses is vital, as the forced shift to a 100% digital work and life context has significantly impacted web survey responses. The number of webs survey has increased dramatically, and various approaches have been applied [39]. In this context, these findings suggest that participants are more likely to complete a web survey after the experience of remote teaching during the pandemic and the familiarity they experienced with these tools [39,40].

## 9. Implications and Limitations

One implication of these results for achieving higher response rates and improving the quality of web surveys, in general, maybe those researchers would be well advised to seek out sample population groups with a particular interest in the survey topic. Researchers can also seek the help of large organizations or prestigious individuals to share the survey. Moreover, an introductory text on the cover page with details about the research's purpose and how their responses would support this purpose is likely to increase the response rate of the web survey. Additionally, the web survey structure/form should be paid attention to during the planning of the web survey. Teachers are more likely to complete a questionnaire if the questions are simple and comprehensive, close-ended, and the questionnaire requires a logical time such as up to ten minutes.

Moreover, to reduce teachers' embarrassment to questions they do not desire to answer, we must include the option "I do not know and, or I do not answer" and place the demographic information at the beginning of the web questionnaire. Our findings indicated that researchers should pay attention to teachers' data safety, privacy, and their consent for participation. Likewise, the research planning should be checked by an official committee responsible for the research's ethical approval. Demographic information such as email addresses and the city of residence should be avoided.

This study has some limitations that could serve as avenues for further research. Due to the use of cross-sectional data, we should be cautious about the inferences about causality [32]. Also, the fact that volunteers were asked to present their views are issues that usually lead to response biases [41]. Another limitation of this research is the almost somewhat limited number of respondents. Although we sent invitations to 400 teachers, only 263 of them completed the survey. However, we submit that this sample is not negligible since, in Greece, similar surveys that study the response rate on web surveys with a sample of teachers have not been conducted. Another limitation may be that no reminder was sent to participants to complete the survey. Suggestions for future research

using mixed methods could carry out the same survey on a larger sample of teachers so that the results can be more generalizable.

Moreover, this research instrument could be used with teachers in other countries to identify possible similarities or differences. Afterwards, could be considered other demographics of participants, such as their various personality traits, may influence the teacher's intention to participate in a web survey with a questionnaire. In addition, it would be advisable to investigate further factors such as "authority" and "number of reminders", as in the relevant literature, the results seem contradictory.

Finally, considering the results of our research regarding the factors that influence someone to participate in a web survey, we suggest a list of recommendations that a researcher should keep in mind when designing a web survey to prevent low response rates:

- Target a sample whose interests are relevant to the subject of the web survey
- Disclose to the respondent the information of the organization or the person conducting the web survey so that they know it is a prestigious body
- State in the invitation the exact purpose of the web survey, the reasons that the teacher should complete the questionnaire, as well as you should indicate the way to inform them about the results
- Indicate to the respondents the approval of the research from the ethics committee
- Ask for participant's consent before starting the web survey
- The questions should be formulated in a straightforward manner
- I prefer closed-ended questions and insert the option "I do not know/I do not answer".
- Indicate the required time to complete the web survey and the number of questions it contains. The web survey should not take more than 10 min to complete
- The timeframe for completion should be about three weeks
- Demographic information should be included at the beginning of the web questionnaire. However, you should avoid asking for the email and city residence of the respondents
- Send the invitation for participation in the web survey by email using the Blind Carbon Copy (BCC) approach.

**Author Contributions:** Data curation, K.L.; Formal analysis, K.L.; Methodology, K.L. and V.K.; Project administration, V.K., A.J. and V.G.; Resources, K.L., A.P. and S.P.; Software, Z.A.; Supervision, V.G.; Validation, A.P.; Visualization, A.P.; Writing–original draft, S.P. and Z.A. All authors have read and agreed to the published version of the manuscript.

**Funding:** This research received no external funding.

**Institutional Review Board Statement:** The research protocol conforms to the ethical guidelines of the European Union. Permission was given by teachers prior to the research. Ethical considerations and guidelines concerning the privacy of individuals were carefully taken into account throughout the whole research process (https://research.upatras.gr/portfolio/ehe/?lang=en accessed on 1 June 2022).

**Informed Consent Statement:** Informed consent was obtained from all subjects involved in the study.

**Data Availability Statement:** The data are available after conducting the corresponding author.

**Conflicts of Interest:** The authors declare no conflict of interest.

## Appendix A

Questionnaire

1. I will participate in a web survey with a questionnaire (Strongly Disagree to Agree Strongly)
2. I complete research conducted by prestigious organizations (Universities, Government agencies, public agencies, etc.) (Strongly Disagree to Agree Strongly)
3. I fill out a web questionnaire conducted by an organization I belong to (or if my institution sponsored the survey) (Strongly Disagree to Agree Strongly)

4. I fill out a web questionnaire conducted by a person I know (e.g., teacher, colleague, friend) (Strongly Disagree to Agree Strongly)
5. I participate in a survey with a web questionnaire if only I am promised a monetary reward (Strongly Disagree to Agree Strongly)
6. I participate in a survey with a web questionnaire if I am aware that my participation will offer an amount as a donation to an organization (Strongly Disagree to Agree Strongly)
7. I participate in a web survey to benefit society by helping inform research; I generally see value in research efforts (Strongly Disagree to Agree Strongly)
8. I complete a survey with a web questionnaire if I am interested in the topic (Strongly Disagree to Agree Strongly)
9. I participated in a survey with a web questionnaire if the researcher promised to inform me about the findings (Strongly Disagree to Agree Strongly)
10. I participate in a web survey with a questionnaire when the subject indicates the research purpose (Strongly Disagree to Agree Strongly)
11. I fill out a web questionnaire when the answers are not mandatory (Strongly Disagree to Agree Strongly)
12. I participate in a survey with a web questionnaire if I know how long it will take to fill out beforehand (Strongly Disagree to Agree Strongly)
13. I participate in a survey with a web questionnaire if I know the number of questions (Strongly Disagree to Agree Strongly)
14. I participate in a web survey with a questionnaire if the invitation includes my name (Strongly Disagree to Agree Strongly)
15. If the email invitation looks professional, I participate in a web survey with a questionnaire (Strongly Disagree to Agree Strongly)
16. I participate in a web survey with a questionnaire if only the questions items are petite and summarizing (Strongly Disagree to Agree Strongly)
17. I participate in a web survey with a questionnaire if only the question items are open-ended (not all possible answers are included, and text is needed) (Strongly Disagree to Strongly Agree)
18. I participate in a web survey with a questionnaire if only the questions are closed-ended (including all possible answers) (Strongly Disagree to Agree Strongly)
19. I participated in a web survey with a questionnaire if only the questions included "I do not know and, or I do not answer" options (Strongly Disagree to Agree Strongly)
20. The best required time to complete a questionnaire is: less than 5′, 6–10′, 11–15′, 16+
21. I participate in a web survey with a questionnaire if the deadline is: 1 week, two weeks, three weeks, or four weeks.
22. The best place for demographic information in a questionnaire is: at the beginning, at the end, not to exist.
23. I fill out a web questionnaire when I am asked about my consent (Strongly Disagree to Agree Strongly)
24. I participate in web surveys with a questionnaire if I am assured of my anonymity (Strongly Disagree to Agree Strongly)
25. I participate in web surveys with a questionnaire that I know secures my data and information (Strongly Disagree to Agree Strongly)
26. I would like to know how the researcher accessed my account when the registration invitation was sent via email (Strongly Disagree to Agree Strongly)
27. I participate in a web survey with a questionnaire if this research has received the approval of the research ethics committee (Strongly Disagree to Agree Strongly)
28. What is the demographic information that could discourage you from participating in the web survey (open-ended question)
29. I participate in a web survey with a questionnaire when the invitation to participate is sent by email (Strongly Disagree to Agree Strongly)
30. I participate in a web survey with a questionnaire; if I see on a website the invitation (link) (Strongly Disagree to Agree Strongly)

31. I participate in a web survey with a questionnaire if only I receive a reminder to complete it (Strongly Disagree to Agree Strongly)

Stability of the research instrument

**Table A1.** Wilcoxon signed-rank nonparametric test for a test-retest process (N = 15).

|  | Z | p |
| --- | --- | --- |
| q1_post-q1_pre | −0.378 | 0.705 |
| q2_post-q2_pre | −0.378 | 0.705 |
| q3_post-q3_pre | −1.414 | 0.157 |
| q4_post-q4_pre | −1.732 | 0.083 |
| q5_post-q5_pre | −0.333 | 0.739 |
| q6_post-q6_pre | −0.816 | 0.414 |
| q7_post-q7_pre | −0.577 | 0.564 |
| q8_post-q8_pre | −1.414 | 0.157 |
| q9_post-q9_pre | −1.000 | 0.317 |
| q10_post-q10_pre | −0.816 | 0.414 |
| q11_post-q11_pre | −0.577 | 0.564 |
| q12_post-q12_pre | −1.414 | 0.157 |
| q13_post-q13_pre | −0.378 | 0.705 |
| q14_post-q14_pre | −1.300 | 0.194 |
| q15_post-q15_pre | −1.633 | 0.102 |
| q16_post-q16_pre | −1.823 | 0.068 |
| q17_post-q17_pre | −1.000 | 0.317 |
| q18_post-q18_pre | −1.342 | 0.180 |
| q19_post-q19_pre | −1.414 | 0.157 |
| q20_post-q20_pre | −0.447 | 0.655 |
| q21_post-q21_pre | −1.035 | 0.301 |
| q22_post-q22_pre | −1.000 | 0.317 |
| q23_post-q23_pre | −1.414 | 0.157 |
| q24_post-q24_pre | −1.000 | 0.317 |
| q25_post-q25_pre | −1.732 | 0.083 |
| q26_post-q26_pre | −1.000 | 0.317 |
| q27_post-q27_pre | −0.577 | 0.564 |
| q29_post-q29_pre | −1.289 | 0.197 |
| q30_post-q30_pre | −1.265 | 0.206 |
| q31_post-q31_pre | −1.000 | 0.317 |

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
