# Peer review of "Factors Affecting Response Rates of the Web Survey with Teachers"

_computers, doi:10.3390/computers11090127_

Round 1
Reviewer 1 Report
The comments on this manuscript are listed as follows.
1. The authors mentioned some factors had been studied scarcely from the previous research in the first paragraph of page 2; they should explain how did they know such factors had not been studied scarcely in the previous research.
2. The authors explained the content of their designed questionnaire in the 3rd paragraph of page 4. The authors described that there are 31 items in the second section of the questionnaire; however, the sum of groups of items, authority (3 items), incentives (6 items), survey structure/form (12 items), ethical issues (6 items) and reminders, and pre-notification (3 items) is 30, not 31.
3. The authors should explain the source or basis of all the items in the questionnaire.
4. On page 4, the authors described the overall response rate of the questionnaire in this study is 65.75%. In this study, the authors should explore this response rate further.
5. The “Per cent” in Table 1 should be “Percent”; moreover, the decimal symbol in the “Percent” column should be ‘.’, instead of ‘,’.
6. The authors described that a 4-point Likert-type scale (1. strongly disagree, 2. disagree, 3. Agree, and 4. strongly agree) was adopted in this study; however, Table 2~6 show the “SD to D” and “A to SA” columns. The authors should explain the meanings of the two columns, “SD to D” and “A to SA”, in detail.
7. The legends in Table 2~6 should explain all the abbreviations.
8. The sum of the items in Table 2~6 is 26. And, the authors mentioned that the questionnaire has 31 items. The authors should explain why only 26 items are shown in Table 2~6. Moreover, it had better for the authors to list the contents of the questionnaire in the manuscript.
9. Figure 1 shows that “email address” discourages participants from completing a web survey; however, in the “Discussion of results” section, the authors described that participants are more likely to complete a web survey when the invitation to participate is sent by email. There exists a controversy. The authors should explain this controversy further.
10. The authors should check the correction of all literature citations in the manuscript.
Author Response
Dear colleague, we would like to thank you for your valuable comments, which helped us to improve the paper further. We will find the revised manuscript and the necessary documents in the attached file, including our answers-response to your comments.
With regards,

Reviewer 2 Report
The submitted paper addressed an interesting methodological issue: the response rates in web surveys. The topic is well argumented and documented with previous studies from the scientific literature.
There is a very good connection between previous literature and the elements included in the questionnaire. Every aspect considered (authority, incentives, survey structure/form, ethical issues, reminders and pre-notifications, and survey time) is identified in the literature review and, after that, is included in the research instrument.
The use of the Wilcoxon test to test the stability of the instrument is to be appreciated but for accuracy of reporting, the exact test value and significance threshold (p) should be included (last paragraph on page 4)
For The strategy of data analysis, I have one recommendation, namely the inclusion of a literature reference that supports that 0.2 is a good correlation in the case of Somers'd (last paragraph before Participants)
Regarding “Pearson chi-square results do not indicate significant associations (p>0.05) between the teachers' intention and all the demographic variables: gender, group of age, educational stage, postgraduate studies, and research experience of teachers” (page 6), the authors must include the value of chi-square for each association and the significance level as well.
The results are presented in a logical and clear manner which makes the paper easy to read and understand.
Author Response

(The authors gave the same response as above.)

Round 2
Reviewer 1 Report
1. The “Per cent” still appeared in Table 1; it should be “Percent”.
2. It had better for the authors to keep the explanation of ‘A’ and ‘D’ in the legends of Table 2 ~ Table 6.
Author Response
Dear reviewer, based n your 2nd round comments, we further revised the paper. You will see the paper based on your comments in the attached file.
Thank you for your efforts to help us improve our paper further.
With regards,
Our response to your comments:
- The “Per cent” still appeared in Table 1; it should be “Percent”.
We revised this word.
- It had better for the authors to keep the explanation of ‘A’ and ‘D’ in the legends of Table 2 ~ Table 6.
We added explanations for the A and D in all the tables from 2 to 6.
